# OpenReview forum: "On the Resilience of Multi-Agent Systems with Malicious Agents"
_ICLR.cc/2025/Conference — Submitted to ICLR 2025_

### Official Review · Reviewer_3j9A · 2024-10-26

**Soundness:** 3
**Presentation:** 3
**Contribution:** 2
**Rating:** 5
**Confidence:** 4

**Summary:**

The paper proposes to investigate the case of LLM-based agents that are collaborating on a task (such as a coding task), when some of the agents are malicious. The paper considers several patterns of communication between the agents such as A->B->C or A<->B<->C.

The authors propose methods to transform agents into malicious ones. These methods essentially involve prompts that tell the agents to introduce subtle errors into their output, such as code.

Investigating various communication architectures, they argue that hierarchical architectures are less vulnerable to malicious agents compared to flat and linear ones. The paper also introduces countermeasures, such as agents that challenge the malicious output, and asks the malicious agent to correct its output.

**Strengths:**

* The problem of malicious LLM-based agents participating in tasks is an important topic.
* The paper shows the results of a relatively extensive experimentation and development of prompts for agent modeling.
* The authors found several interesting and non-trivial insights into the behavior of certain LLM agent models. Some of these are the introduction of errors can improve the output of agent models that are based on debates.

**Weaknesses:**

* The paper seem to be unaware of the existing, and very well known literature of malicious agents in a system (such as the Byzantine Generals problem in its many variations). There are algorithms that are extensively used in networking protocols and database systems.
* The paper presents as new discoveries facts such as hierarchical systems are more resilient because the agent at the top of the hierarchy is provided "with various versions of the answer by multiple agents performing the same sub-task". This is not a property of hierarchy, but of replication - again, distributed system theory contains many algorithms that can show how to protect against malicious agents in a fully flat and distributed environment.
* The various agent implementations considered in this paper are essentially relatively short prompts provided to ChatGPT. The validity of various observations is thus dependent on the current version of ChatGPT, which might be different by the time this paper is presented.
* Some of the observations are also dependent on the limitations of current LLMs - for instance, the observation that the malicious agents gradually loose track of the assignment to introduce errors. These are problems that can be easily fixed by periodically reintroducing the tasks.

**Questions:**

* Do you expect that the observations in this paper about the relative strengths of different architectures will be still valid for the next versions of language models? What happens if this paper is published and becomes part of the knowledge-base of the LLMs?
* The agents (even the malicious ones) do not seem to be aware of the architecture of the overall system. Does this matter?

**Details Of Ethics Concerns:**

The paper contains source code for the prompt for a malicious agent that tries to deceive the user about it maliciousness. Overall, the impact of such released source code is minimal, because examples of such prompts are widely available. The objective of the paper it to minimize the impact of such malicious agents, a legitimate research problem.

Overall, I believe that this should not impact the paper, but it can benefit from the insight of an ethics reviewer.

---

> ### Author Response · Authors · 2024-11-25
> **Official Response (1/n)**
>
> We deeply appreciate your efforts in reviewing and recognition of our experiment’s comprehensiveness and non-trivial insights. Your feedback has significantly improved our paper. In the response, we address your concerns one by one.
>
> > The paper seem to be unaware of the existing, and very well known literature of malicious agents in a system (such as the Byzantine Generals problem in its many variations). There are algorithms that are extensively used in networking protocols and database systems.
>
> Thank you for your insightful comment. We acknowledge that the Byzantine Generals problem and its variations represent foundational work in understanding malicious agents in distributed systems, with numerous algorithms addressing attacks such as DDoS, MITM, Sybil, and impersonation [3, 4, 5]. While these studies focus on **designing specific system-level mechanisms like authentication, authorization, and confidentiality** [1, 2, 6], our work diverges by exploring **the organizational structures and interaction strategies of LLM-based multi-agent systems**. The scenarios we investigate **mirror real-world human collaboration dynamics rather than the behavior of traditional distributed nodes**, offering a novel perspective distinct from conventional solutions.
>
> [1] Reiter, Michael, Kenneth Birman, and Li Gong. "Integrating Security in a Group Oriented Distributed System." Proceedings 1992 IEEE Computer Society Symposium on Research in Security and Privacy. IEEE Computer Society, 1992.
>
> [2] Satyanarayanan, Mahadev. "Integrating security in a large distributed system." ACM Transactions on Computer Systems (TOCS) 7.3 (1989): 247-280.
>
> [3] Harinath, Depavath, P. Satyanarayana, and M. R. Murthy. "A review on security issues and attacks in distributed systems." Journal of Advances in Information Technology 8.1 (2017).
>
> [4] Brown, Philip N., Holly P. Borowski, and Jason R. Marden. "Security against impersonation attacks in distributed systems." IEEE Transactions on Control of Network Systems 6.1 (2018): 440-450.
>
> [5] Kumar, Manoj, and Nikhil Agrawal. "Analysis of different security issues and attacks in distributed system a-review." International Journal of Advanced Research in Computer Science and Software Engineering 3.4 (2013): 232-237.
>
> [6] Mudholkar, P. K., and M. Mudholkar. "Security in distributed system." Proceedings of the International Conference and Workshop on Emerging Trends in Technology. 2010.
>
> > The paper presents as new discoveries facts such as hierarchical systems are more resilient because the agent at the top of the hierarchy is provided "with various versions of the answer by multiple agents performing the same sub-task". This is not a property of hierarchy, but of replication - again, distributed system theory contains many algorithms that can show how to protect against malicious agents in a fully flat and distributed environment.
>
> Thank you for this insightful comment. While replication indeed enhances resilience, our argument highlights the additional role of hierarchy in improving performance. For instance, in the MAD system, **removing the Judge transforms the structure into a flat configuration where two agents debate directly**. Although replication ensures multiple interaction rounds, **the absence of hierarchical oversight degrades performance**, as the Judge's role in aggregating and adjudicating inputs is critical for efficiency and accuracy. This distinction underscores the unique contribution of hierarchical systems beyond simple replication.
>
> > The various agent implementations considered in this paper are essentially relatively short prompts provided to ChatGPT. The validity of various observations is thus dependent on the current version of ChatGPT, which might be different by the time this paper is presented.
> > Do you expect that the observations in this paper about the relative strengths of different architectures will be still valid for the next versions of language models? What happens if this paper is published and becomes part of the knowledge-base of the LLMs?
>
> Thank you for raising this important point. We acknowledge that advancements in LLMs may impact performance. To address this, we conducted experiments with different iterations of GPT, specifically GPT-3.5 and GPT-4o, **as detailed in Appendix A (Line 772, Page 15).** The results show a general performance improvement with GPT-4o while our **core conclusions remain consistent**: hierarchical structures consistently outperform others, rigorous tasks are more susceptible to malicious agents, and systems like MAD and Camel also exhibit performance gains. This suggests that **our findings are robust across model updates**, providing a strong foundation for future iterations of LLMs.

---

> > ### Author Response · Authors · 2024-11-25
> > **Official Response (2/n)**
> >
> > > Some of the observations are also dependent on the limitations of current LLMs - for instance, the observation that the malicious agents gradually loose track of the assignment to introduce errors. These are problems that can be easily fixed by periodically reintroducing the tasks.
> >
> > Thank you for this insightful comment. While periodically reintroducing tasks could address issues of task drift in malicious agents, our primary goal with AutoTransform is to implement **a one-time modification of agent profiles** without ongoing intervention. **Incorporating periodic task reintroduction would require significant modifications to the framework** (e.g., appending tasks to the latest user prompt), which falls outside the scope of our intended methodology.
> >
> > > The agents (even the malicious ones) do not seem to be aware of the architecture of the overall system. Does this matter?
> >
> > Thank you for the insightful observation. We analyzed six systems to address this point. In **Camel, MAD, and SPP**, agents are aware of the overall system architecture, while in **Self-collab, MetaGPT, and AgentVerse**, they are not. Despite this difference, the performance drop under AutoInject is comparable: from 63.2 to 39.3 for architecture-aware systems and from 66.3 to 39.0 for architecture-unaware systems, as shown below:
> >
> > | | Aware | Unaware |
> > |---|---|---|
> > | Vanilla | 63.2 | 66.3 |
> > | AutoInject | 39.3 | 39.0 |
> >
> > Currently, **our attack design does not leverage architectural knowledge** (e.g., instructing agents to target specific recipients). This is an intriguing area for future exploration and could further elucidate the impact of architecture awareness on system robustness.
> >
> > > The paper contains source code for the prompt for a malicious agent that tries to deceive the user about it maliciousness. Overall, the impact of such released source code is minimal, because examples of such prompts are widely available. The objective of the paper it to minimize the impact of such malicious agents, a legitimate research problem. Overall, I believe that this should not impact the paper, but it can benefit from the insight of an ethics reviewer.
> >
> > Thank you for your thoughtful feedback. We acknowledge the potential misuse of the prompts described in our paper. To address this concern, we have proposed and rigorously evaluated two defense methods—**Challenger and Inspector—and their combination**, which are specifically designed to mitigate the influence of malicious agents. The results, summarized in the tables below, demonstrate the effectiveness of these defenses across different scenarios and attack types:
> >
> > | Self-collab | No Defense | Challenger | Inspector | Challenger + Inspector |
> > |---|---|---|---|---|
> > | No Attack | 76.22 | 74.56 | 76.39 | 76.83 |
> > | AutoTransform | 43.29 | 70.73 | 74.40 | 75.00 |
> > | AutoInject | 40.85 | 71.95 | 67.68 | 73.78 |
> >
> > | Camel | No Defense | Challenger | Inspector | Challenger + Inspector |
> > |---|---|---|---|---|
> > | No Attack | 62.20 | 62.23 | 61.03 | 63.79 |
> > | AutoTransform | 32.46 | 43.50 | 41.75 | 48.70 |
> > | AutoInject | 29.27 | 40.24 | 44.16 | 48.64 |
> >
> > These defenses significantly reduce the effectiveness of malicious prompts while preserving system performance under benign conditions. Additionally, we agree that an ethics review could further enhance the paper's insights, and we welcome the input of an ethics reviewer if deemed necessary.

---

> > > ### Comment · Reviewer_3j9A · 2024-11-28
> > >
> > > The author's responses do not change my original evaluation of the paper.

---

> > > > ### Author Response · Authors · 2024-11-28
> > > >
> > > > We appreciate your time for reading our response. We are glad to further address your concerns. Please feel free to reach out.

---

### Official Review · Reviewer_54qu · 2024-11-02

**Soundness:** 3
**Presentation:** 3
**Contribution:** 3
**Rating:** 8
**Confidence:** 3

**Summary:**

The paper explores the resilience of MAS comprising agents with LLM in the presence of malicious agents. The authors focus on evaluating the robustness of different MAS structures—linear, flat, and hierarchical—across tasks such as code generation, math problem solving, translation, and text evaluation. To simulate malicious agent behavior, two approaches—AUTOTRANSFORM and AUTOINJECT—are introduced. The study's findings indicate that hierarchical structures show superior resilience, with additional strategies for enhancing system robustness, including the "Challenger" and "Inspector" defense mechanisms.

**Strengths:**

Solid Experimentation: The experimental setup is robust, involving several MAS architectures and tasks, and employs quantitative measures that provide a detailed analysis of resilience across scenarios.

Insightful Findings on Architecture Impact: The conclusion that hierarchical systems exhibit better resilience, supported by performance metrics, is particularly valuable for MAS design in practical applications where security and reliability are critical.

Contributions to LLM Safety: The exploration of malicious agent effects and subsequent defenses offers important insights into enhancing MAS reliability, especially in decentralized or unregulated environments.

**Weaknesses:**

1. The experiment models are limited: As it only tests on the gpt-based models, gpt3.5 and gpt4o.
2. The paper presents results across tasks that involve different cognitive demands (e.g., code generation requiring precision versus translation being more subjective). However, there is limited analysis of how the degree of agent specialization affects system resilience in different MAS structures.
3. While the chosen tasks (code generation, math, translation, and text evaluation) provide a reasonable testbed, they may not fully represent the diversity of tasks that MAS are deployed to handle. These tasks are fairly discrete and objective; however, multi-agent systems in more nuanced, real-world applications (e.g., recommendation engines or dynamic response systems) might face unique types of malicious behavior. Including a more diverse array of tasks or explaining the rationale behind the current selection would strengthen the applicability of the findings.
4. While the paper explores the impact of error rates (Pm and Pe), the analysis remains somewhat superficial. It lacks a nuanced discussion of why certain error rates were chosen and how these rates might affect system resilience in different real-world applications.

**Questions:**

1. Following the weakness point 2, whether agents with specialized roles (e.g., a math-focused agent vs. a generalist) exhibit varying vulnerabilities to malicious behaviors is not fully explored. This is a missed opportunity to highlight if specialized roles within the MAS require additional security considerations or different structural adjustments.
2. Following the weakness point 1, I suggest you can deploy experiments on the o1-mini, o1-preview, I hope to see the results.
3. For weakness point 3 and 4, I hope you can improve it in your next exploration in this topic.

---

> ### Author Response · Authors · 2024-11-25
> **Official Response (1/n)**
>
> We deeply appreciate your efforts in reviewing and recognition of our experiment’s comprehensiveness and impactful insights. Your feedback has significantly improved our paper. In the response, we address your concerns one by one.
>
> > The experiment models are limited: As it only tests on the gpt-based models, gpt3.5 and gpt4o. Following the weakness point 1, I suggest you can deploy experiments on the o1-mini, o1-preview, I hope to see the results.
>
> Thank you for the valuable suggestion. To broaden the scope of our investigation and ensure our conclusions generalize **beyond the GPT model family**, we conducted additional experiments using one of **the state-of-the-art open-source models, the LLaMA-3.1-70B-Instruct**. The results, presented in the tables below, confirm that our findings hold across diverse model architectures, including non-GPT-based LLMs.
>
> | LLaMA-3.1-70B-Instruct | Linear | Flat | Hierarchical |
> |---|---|---|---|
> | No Attack | 73.78 | 76.83 | 76.15 |
> | AutoTransform | 11.90 | 39.03 | 66.96 |
> | AutoInject | 38.72 | 36.59 | 55.64 |
>
> > The paper presents results across tasks that involve different cognitive demands (e.g., code generation requiring precision versus translation being more subjective). However, there is limited analysis of how the degree of agent specialization affects system resilience in different MAS structures. Following the weakness point 2, whether agents with specialized roles (e.g., a math-focused agent vs. a generalist) exhibit varying vulnerabilities to malicious behaviors is not fully explored. This is a missed opportunity to highlight if specialized roles within the MAS require additional security considerations or different structural adjustments.
>
> Thank you for this insightful comment. **In Section 4.7 (Line 422, Page 8)**, we conducted an initial experiment where the Manager (Instructor) was made malicious instead of the Coder in Camel and MetaGPT. Our findings indicate that **compromising higher-level task distributors leads to a more significant performance decline** in both systems. While this provides preliminary insights, we acknowledge the need for a more comprehensive analysis of how agent specialization impacts resilience to malicious behaviors. We have noted this as an important avenue for future research, as the primary focus of this paper is on the influence of organizational structures in MAS.
>
> > While the chosen tasks (code generation, math, translation, and text evaluation) provide a reasonable testbed, they may not fully represent the diversity of tasks that MAS are deployed to handle. These tasks are fairly discrete and objective; however, multi-agent systems in more nuanced, real-world applications (e.g., recommendation engines or dynamic response systems) might face unique types of malicious behavior. Including a more diverse array of tasks or explaining the rationale behind the current selection would strengthen the applicability of the findings.
>
> Thank you for this insightful comment. We acknowledge that the four tasks chosen may not fully capture the diversity of real-world multi-agent system applications. Our selection was guided by **their prevalence in existing literature and their suitability for evaluating the resilience of the six systems studied**. We recognize the importance of exploring more nuanced and dynamic scenarios, such as recommendation systems or multidisciplinary team consultations in healthcare, and will incorporate these in future research to enhance the applicability of our findings.

---

> > ### Author Response · Authors · 2024-11-25
> > **Official Response (2/n)**
> >
> > > While the paper explores the impact of error rates (Pm and Pe), the analysis remains somewhat superficial. It lacks a nuanced discussion of why certain error rates were chosen and how these rates might affect system resilience in different real-world applications.
> >
> > We appreciate the reviewer’s observation regarding the need for a nuanced discussion on error rates and their real-world implications. While we acknowledge that different error types can have varied impacts on program functionality and detection difficulty, we selected line of code (LOC) as a metric for **its simplicity and widespread acceptance in software development**. Although more sophisticated metrics exist, LOC remains a standard and practical measure for assessing program complexity and error distribution.
> >
> > To address the concern further, we **categorized and quantified the errors introduced in our analysis into seven distinct types**, as shown in the table below. This categorization ensures diversity in error representation, thereby enhancing the robustness of our results. By generating a large number of errors across these categories, we aim to mitigate biases introduced by any single error type and provide a comprehensive evaluation of system resilience.
> >
> > | Category Name | Description | Count |
> > |---|---|---|
> > | Logical Errors | Errors in logical operations, such as incorrect operators or inverted logic. | 12 |
> > | Indexing and Range Errors | Issues with boundary conditions or off-by-one indexing. | 23 |
> > | Mathematical Errors | Errors in calculations or numerical processing. | 20 |
> > | Output and Formatting | Issues with producing or formatting expected output. | 9 |
> > | Initialization Errors | Problems with starting values or incorrect initialization. | 4 |
> > | Infinite Loops | Errors causing unintended infinite execution loops. | 6 |
> > | Runtime Invocation Issues | Errors in function calls or runtime handling. | 6 |
> >
> > This categorization supports our conclusion that the **diverse error set adequately covers a range of scenarios**, allowing for meaningful evaluation of system behavior under varying conditions.

---

> > > ### Comment · Reviewer_54qu · 2024-11-26
> > > **Thank you for the reply**
> > >
> > > Your answers solve my concerns. I have raised my score to 8.

---

> > > > ### Author Response · Authors · 2024-11-27
> > > >
> > > > Thank you very much for reading our responses. We deeply appreciate your recognition!

---

### Official Review · Reviewer_jrv1 · 2024-11-02

**Soundness:** 2
**Presentation:** 3
**Contribution:** 3
**Rating:** 6
**Confidence:** 4

**Summary:**

This paper studies the resilience of multi-agent systems (with LLM agents) to the introduction of errors by malicious agents. The authors consider two methods for introducing such errors ("AutoTransform" and "AutoInject") and two methods for mitigating them (an "Inspector" and "Challenger" agent). They study the impact of these methods on three multi-system structures (linear, flat, and hierarchical) in the context of four downstream tasks (code generation, maths, translation, and text evaluation). Overall, they find that hierarchical structures are more resistant to the introduction of errors, that such errors can actually _aid_ performance in some settings, and that their strategies for mitigating the introduction errors are helpful. They also study other questions such as the impact of the type vs. rate of errors, etc.

**Strengths:**

This is an important and increasingly relevant topic, and the paper does a good job of outlining some interesting research questions in the area. The authors also make reasonable choices of systems and tasks to study. Overall I found the paper well-structured and relatively easy to read (there are a few minor typos here and there, but that didn't affect the scientific quality and clarity of the paper). I appreciated the clear statement of the research questions especially. I thought their experiments seemed mostly well-designed, and their choices of methods (both for the introduction of errors and for defending against them) were sensible.

**Weaknesses:**

The main weaknesses of the paper, in my opinion, are two-fold.

First and most importantly, it is not always clear exactly what the stated results represent and how significant they are:

- In several places it is not clear why the authors test some experimental configurations but not others. These absences may well be justifiable, but the authors should provide clear justification (and otherwise include the additional configurations, if only in the appendices). For example:
   - In Figure 4 the authors only evaluate MAD.
   - MAD is not evaluated in Figure 7a.
   - In Figure 8 the authors only evaluate Camel.
- What does "Vanilla" mean in Figure 3 and elsewhere? In Figure 2 it seems as though the idea is that one agent is responsible for repeating(?) the task description and another for executing the task, but I assume it cannot be this simple. How does it generalise when there are more than two agents?
- In several tables and bar charts it is not always clear what tasks are actually being evaluated or how much variation there is between these tasks. E.g. in Figure 5 it simply says "selected downstream tasks".
- There are no error bars or standard errors reported anywhere, which makes it difficult to interpret the statistical significance of the results.

Secondly, and less importantly, I found that some of the claims the authors made seemed overly strong, and that they were excessively focused on the context of LLMs, despite the vast literature on fault-tolerance in multi-agent systems more generally. I suggest that the authors caveat their claims appropriately and aim to discuss how their work results to similar efforts in the context of non-LLM agents. As specific examples:

- In one instance, the authors claim that they are "the first to examine how different structures of multi-agent systems affect resilience" to malicious agents, which is clearly false in general (as opposed to the special case of LLM agents). Indeed, as far as I can tell, none of the literature on game theory and the fault-tolerance of multi-agent/distributed systems is cited in the related work section. I suspect that there are many ideas in that literature that the authors might find useful for solving the problems they are interested in.
- Relatedly, even when restricting to the LLM setting, I recently came across (but have not yet read in full) the paper "Prompt Infection: LLM-to-LLM Prompt Injection within Multi-Agent Systems" (arXiv:2410.07283), which seems closely related to this work. To the extent that it is, the authors of this paper should comment on the differences and similarities, including to any other relevant work that is referenced within the "Prompt Infection" paper.
- As a small point, in line 156 it is claimed that it is not possible to use LLMs to inject syntax errors in 20% of the lines in some code, but I am somewhat suspicious of this assertion. Having personally used LLMs for very similar tasks in the past, it is my impression that SOTA models are can do a reasonable job of following such instructions (especially when combined with additional checks applied to the resulting code).

Finally, I noticed in Appendix B.2 that the prompt for the text evaluation problem explicitly tells the agent which model generated which text (ChatGPT or Vicuna-13B). For the evaluation to be unbiased, surely the model outputs should be anonymised?

**Questions:**

Please see the Weaknesses section for my questions. I also welcome the authors to correct any misunderstandings I may have about their paper.

As an additional note to the authors, I have currently selected "marginally below the acceptance threshold" but I believe most of the weaknesses above are addressable without too much additional effort. If that were to be done (and the paper updated accordingly within the rebuttal period) I would happily increase my score in order to recommend acceptance.

**Details Of Ethics Concerns:**

This paper proposes novel methods for adversarially attacking multi-agent LLM systems. I am uncertain of whether this meets the bar for requiring an ethics review, but it is clearly relevant to the privacy, security, and safety of AI systems.

---

> ### Author Response · Authors · 2024-11-25
> **Official Response (1/n)**
>
> We deeply appreciate your efforts in reviewing and your recognition of the importance of the research questions and the clear presentation of our paper. Your feedback has significantly improved our paper. In the response, we address your concerns one by one.
>
> > In several places it is not clear why the authors test some experimental configurations but not others. These absences may well be justifiable, but the authors should provide clear justification (and otherwise include the additional configurations, if only in the appendices). For example: In Figure 4 the authors only evaluate MAD. MAD is not evaluated in Figure 7a. In Figure 8 the authors only evaluate Camel.
> > In several tables and bar charts it is not always clear what tasks are actually being evaluated or how much variation there is between these tasks. E.g. in Figure 5 it simply says "selected downstream tasks".
>
> We appreciate the reviewer’s thoughtful feedback. Below, we clarify the rationale for our experimental configurations and address the specific concerns raised:
>
> - Figure 4 (MAD): This figure focuses on **a case study** demonstrating a counter-intuitive phenomenon where introducing errors can improve performance—a rare observation in multi-agent systems. MAD was selected specifically for its relevance to this unique insight.
> - Figure 7a (Exclusion of MAD): MAD was excluded from Figure 7a because this experiment involves scenarios with malicious **instruction-sending agents**, which are not present in the MAD system configuration.
> - Figure 8 (Self-collab and Camel): Only Self-collab and Camel are included in Figure 8 because they **represent the weaker systems within the Linear and Flat structures**, respectively. Our objective in this experiment is to illustrate how our proposed defense method enhances resilience in weaker systems.
>
> To provide greater clarity on our multi-agent system settings, we have added a comprehensive table summarizing the experimental configurations to our presentation, as shown below:
>
> | Systems | Structure | Tasks | N. of Agents | Final Agent | Malicious Agent |
> |---|---|---|---|---|---|
> | MetaGPT | Linear | Code | 5 | Test Engineer | Code Engineer |
> | Self-collab | Linear | Code | 2-5 | Tester | Coder |
> | Camel | Flat | All | 2 | User | Assistant |
> | SPP | Flat | Code | 3 | AI Assistant | Python Programmer |
> | MAD | Hierarchical | All | 3 | Judge | debater |
> | AgentVerse | Hierarchical | All | 4 | Critic | Solver |
>
> > What does "Vanilla" mean in Figure 3 and elsewhere? In Figure 2 it seems as though the idea is that one agent is responsible for repeating(?) the task description and another for executing the task, but I assume it cannot be this simple. How does it generalise when there are more than two agents?
>
> The term "Vanilla" refers to **the baseline scenario where no attack or defense mechanisms are applied**, representing a standard, unmodified system. In Figure 2, it specifically denotes the **normal communication between agents** without any adversarial influence or additional safeguards. This serves as a control setup to evaluate the impact of the proposed methods.
>
> > There are no error bars or standard errors reported anywhere, which makes it difficult to interpret the statistical significance of the results.
>
> Thank you for highlighting this concern. While including error bars or standard errors would indeed provide additional statistical context, conducting repeated experiments for all tasks would incur significant time and resource constraints given the extensive scope of the study. However, the **large number of test cases in each task ensures that our results are statistically robust and representative.**

---

> ### Author Response · Authors · 2024-11-25
> **Official Response (2/n)**
>
> > In one instance, the authors claim that they are "the first to examine how different structures of multi-agent systems affect resilience" to malicious agents, which is clearly false in general (as opposed to the special case of LLM agents). Indeed, as far as I can tell, none of the literature on game theory and the fault-tolerance of multi-agent/distributed systems is cited in the related work section. I suspect that there are many ideas in that literature that the authors might find useful for solving the problems they are interested in.
>
> We appreciate the reviewer’s feedback and the suggestion to consider broader literature. We acknowledge the extensive body of work on the fault-tolerance of distributed systems, including the Byzantine Generals problem and related attacks such as DDoS, MITM, Sybil, and impersonation [3, 4, 5]. However, much of this work focuses on **specific system designs, emphasizing mechanisms like Authentication, Authorization, and Confidentiality** [1, 2, 6].
>
> In contrast, our paper addresses the organizational structures of LLM-based multi-agent systems, which differ significantly in nature. These systems **mimic real-world human collaboration dynamics** rather than functioning as traditional distributed nodes. This novel focus on structural design in LLM-based systems sets our work apart.
>
> [1] Reiter, Michael, Kenneth Birman, and Li Gong. "Integrating Security in a Group Oriented Distributed System." Proceedings 1992 IEEE Computer Society Symposium on Research in Security and Privacy. IEEE Computer Society, 1992.
>
> [2] Satyanarayanan, Mahadev. "Integrating security in a large distributed system." ACM Transactions on Computer Systems (TOCS) 7.3 (1989): 247-280.
>
> [3] Harinath, Depavath, P. Satyanarayana, and M. R. Murthy. "A review on security issues and attacks in distributed systems." Journal of Advances in Information Technology 8.1 (2017).
>
> [4] Brown, Philip N., Holly P. Borowski, and Jason R. Marden. "Security against impersonation attacks in distributed systems." IEEE Transactions on Control of Network Systems 6.1 (2018): 440-450.
>
> [5] Kumar, Manoj, and Nikhil Agrawal. "Analysis of different security issues and attacks in distributed system a-review." International Journal of Advanced Research in Computer Science and Software Engineering 3.4 (2013): 232-237.
>
> [6] Mudholkar, P. K., and M. Mudholkar. "Security in distributed system." Proceedings of the International Conference and Workshop on Emerging Trends in Technology. 2010.
>
> > Relatedly, even when restricting to the LLM setting, I recently came across (but have not yet read in full) the paper "Prompt Infection: LLM-to-LLM Prompt Injection within Multi-Agent Systems" (arXiv:2410.07283), which seems closely related to this work. To the extent that it is, the authors of this paper should comment on the differences and similarities, including to any other relevant work that is referenced within the "Prompt Infection" paper.
>
> We thank the reviewer for bringing this paper to our attention. Although it was uploaded to arXiv after the ICLR submission deadline, we have reviewed it to identify relevant distinctions. Notably, the "Prompt Infection" paper primarily explores **scenarios such as data theft, malware propagation, and social manipulation**, whereas our work focuses on tasks like code generation and mathematical problem solving. Additionally, their study **does not examine how varying organizational structures within multi-agent systems influence outcomes**, which is a key focus of our research.
>
> > As a small point, in line 156 it is claimed that it is not possible to use LLMs to inject syntax errors in 20% of the lines in some code, but I am somewhat suspicious of this assertion. Having personally used LLMs for very similar tasks in the past, it is my impression that SOTA models are can do a reasonable job of following such instructions (especially when combined with additional checks applied to the resulting code).
>
> We appreciate the reviewer’s insight and acknowledge the potential of SOTA LLMs for similar tasks. To clarify, we conducted an analysis using AutoTransform to instruct a GPT-3.5 agent to introduce errors in 20% and 40% of the code lines. The results are summarized below:
>
> | Error Rate | Avg | Std | Min | Max |
> |---|---|---|---|---|
> | Instruct 20% | 1.56 | 3.65 | 0.0 | 14.3 |
> | Instruct 40% | 9.49 | 26.70 | 0.0 | 90.1 |
>
> These results indicate significant variability, with **agents struggling to consistently achieve the precise error rates** of 20% or 40%. This underscores the necessity and robustness of our AutoInject method, which addresses these limitations effectively.

---

> > ### Author Response · Authors · 2024-11-25
> > **Official Response (3/n)**
> >
> > > Finally, I noticed in Appendix B.2 that the prompt for the text evaluation problem explicitly tells the agent which model generated which text (ChatGPT or Vicuna-13B). For the evaluation to be unbiased, surely the model outputs should be anonymised?
> >
> > We appreciate the reviewer’s observation regarding potential bias. To maintain consistency with the design of the dataset and facilitate a direct comparison between our multi-agent results and the single-agent results reported in the original study, we **adhered to the dataset's original prompts**, which include identifying the model source. This approach ensures alignment with prior evaluations and comparability of results.
> >
> > > This paper proposes novel methods for adversarially attacking multi-agent LLM systems. I am uncertain of whether this meets the bar for requiring an ethics review, but it is clearly relevant to the privacy, security, and safety of AI systems.
> >
> > Thank you for highlighting this important concern. We fully acknowledge the potential risks associated with adversarial attacks on multi-agent LLM systems. To address this, we have proposed and evaluated two defense mechanisms: **the Challenger and the Inspector, as well as their combination**. These defenses have been shown to significantly mitigate the influence of malicious agents, as evidenced by the results presented below:
> >
> > | Self-collab | No Defense | Challenger | Inspector | Challenger + Inspector |
> > |---|---|---|---|---|
> > | No Attack | 76.22 | 74.56 | 76.39 | 76.83 |
> > | AutoTransform | 43.29 | 70.73 | 74.40 | 75.00 |
> > | AutoInject | 40.85 | 71.95 | 67.68 | 73.78 |
> >
> > | Camel | No Defense | Challenger | Inspector | Challenger + Inspector |
> > |---|---|---|---|---|
> > | No Attack | 62.20 | 62.23 | 61.03 | 63.79 |
> > | AutoTransform | 32.46 | 43.50 | 41.75 | 48.70 |
> > | AutoInject | 29.27 | 40.24 | 44.16 | 48.64 |
> >
> > These results demonstrate that our methods effectively reduce the success of attacks, improving security and safety in multi-agent LLM interactions. Additionally, the proposed defenses underscore the ethical responsibility to mitigate potential harm.

---

> ### Comment · Reviewer_jrv1 · 2024-11-27
> **Reply to Authors**
>
> I thank the authors for their detailed reply, and I believe that most of my questions and comments have been addressed. I also appreciate the effort they put into running additional experiments. In light of this, I am happy to update my score to recommend acceptance if the authors can add the clarifying remarks in their rebuttal to the actual paper, even if only in the appendices. My questions came about even after a careful reading of the paper, and I noticed that other reviewers had similar questions, so I can only expect that future readers will too. Given that they have already done the hard work of typing the answers out, I would strongly suggest the authors add all of these clarifications to the actual manuscript.
>
> Regarding the fault tolerance of distributed systems, I agree that some of the cited works mentioned are less directly relevant. I had in mind more the (vast) literature that explicitly considers the impact of network topology on robustness. I am no expert on this myself, but I believe this sometimes also falls under the complex systems literature. See, for instance, [this Wikipedia page](https://en.wikipedia.org/wiki/Robustness_of_complex_networks) or [this textbook](https://www.cambridge.org/us/universitypress/subjects/physics/statistical-physics/complex-networks-structure-robustness-and-function?format=HB&isbn=9780521841566). Even some of the very first things one learns in a distributed computing class is that, e.g. star networks can be less robust because they have a single point of failure, etc. Obviously I think it's important to study some of these things in the case of LLMs, but I want to make it clear that I do not view this more fundamental/theoretical aspect underlying the paper to be a novel contribution of the paper (and that is ok, as long as the authors do not claim novelty in this regard).

---

> > ### Author Response · Authors · 2024-11-28
> >
> > We thank you for acknowledging the additional experiments and explanations we provide. We are encouraged that you find these clarifications helpful. These additions significantly enhance our paper's clarity and address potential questions that future readers may have.
> >
> > Regarding fault tolerance in distributed systems, we appreciate the reviewer bringing attention to the broader literature on the impact of network topology on robustness, including references to complex systems. This is indeed a valuable perspective, and we will include this line of work in the revised related work section to provide a more comprehensive discussion and situate our contributions more effectively.

---

### Official Review · Reviewer_Q1ar · 2024-11-03

**Soundness:** 2
**Presentation:** 2
**Contribution:** 2
**Rating:** 3
**Confidence:** 4

**Summary:**

The paper investigates different aspects of the effects of malicious agents on multi-agent systems. Specifically, it investigates four different questions:
1. How do different multi-agent system structures differ in their resilience to malicious agents?
2. How do different tasks differ in their susceptibility to sabotage by malicious agents in multi-agent systems?
3. How do different error rates (and different types of error rates -- rate of messages with errors vs rate of errors per message with errors) differ in their effect on multi-agent systems?
4. How do syntactic and semantic errors differ in their effect on multi-agent systems?

As the backdrop for this investigation, the paper studies three multi-agent structures (linear, flat, and hierarchical) with two instantiations each, applied to four different tasks (code generation, math problem solving, translation with commonsense reasoning, and text evaluation), two types of malicious agent simulation (AutoTransform and AutoInject), and two defense methods (Challenger and Inspector). The experiments are drawn from combining these elements, and the following results are found:

1. Out of the three multi-agent system structures, the hierarchical structure is the most resilient to the malicious agent simulations in the tasks considered;
2. The multi-agent structures studied are less resilient to the malicious agent simulations studied in code generation and math problem solving than translation with commonsense reasoning and text evaluation;
3. Higher error rates generally lead to worse the performance of the multi-agent systems, except that increasing the rate of errors per message with errors beyond 0.4 does not seem to worsen performance. In addition, generally, higher rates of messages with errors is more detrimental to performance than higher rates of errors per message with errors.
4. Semantic errors have a bigger impact on the performance of the multi-agent systems than syntactic errors.

**Strengths:**

**Originality**

The paper seems original and resourceful in its methods for simulating malicious agents. Its hypothesis that the structures of multi-agent systems (linear vs flat vs hierarchical) is central to determining their resilience to malicious agents also seems original and intriguing.

**Quality**

The selection of downstream tasks seems well done for the purposes of covering a wide range of unrelated tasks. The included case studies (section 4.6) were quite interesting, and I'd be excited to see a more systematic assessment of those phenomena and incorporation into the design of ablation experiments. Overall, the justifications presented for various observed phenomena seem coherent and intuitive. I especially liked the discussion on higher rates of errors per message leading to better performance than the middling cases (although observing the chart, the effect seems plausibly negligible).

**Clarity**

I'll argue in the Weaknesses section that the overall setup of the experiments is not particularly clear, but I think this is downstream of the choice of experiments (number of axes of variations and inconsistency in their variations). Given the choice of experiments, the paper was impressively clear and the large number of results are presented in a way that does not overwhelm.

**Significance**

Multi-agent systems seem significant, and their resilience is likely to become a critically important area of research. I commend the authors in their choice of problem. The aspects of the experiments that are analyzed (system structure, tasks, etc) also seem quite relevant for the broader question.

**Weaknesses:**

1. My main concern is that this paper attempts to do too much and is not sufficiently focused. Between the different multi-agent structures, different tasks, different attacks, different types of error rates, different error types, and different defenses, there are too many variables, each investigation ends up with limited depth, and the overall picture ends up not fully compelling. As a result of this ambition, details in the subquestions appear insufficiently investigated. For example, when discussing different multi-agent structures, I hoped to find more details about the structures and their dynamics, and more possible instantiations of each high-level structure (or, more systematic variation in the instantiations considered). Then the rest of the setup could be simplified (for example, it could focus on a single task category such as code generation, again possibly with more than one instantiation of the task category). The resulting claim would have to be more modest -- for example, it would pertain only to code generation and not any task -- but it would be much more strongly substantiated. As it stands (especially given the results do not seem particularly extreme), I'm left wondering if it is really the case that hierarchical structures are more resilient, or if the results apply only to the specific hierarchical systems tested and whether there is something particular about each of them that led to the results. I'm left unconvinced that the main claims of the paper are true.

2. A related general concern is that the lack of systematic ablations or closer analyses of the specific results made me quite doubtful of them. I think many additional experiments could have been very illustrative. For example, in the investigation of the defense methods, I was left wondering how much the relation between the defense methods and the attack methods mattered. I find it plausible that these "defense methods" are just generally useful enhancements to the multi-agent systems, and was interested in seeing results on the multi-agent systems with the "defense methods" even without the attack in place. In that case, the discussion of these results would be a bit different.

3. I found the combination of experiments a bit confusing and unclear. Part of this is directly downstream of point 1 (there are too many axes of variation), but it is also the case that the different factors are, it seems, inconsistently varied. For example, the introduction of "Error Types" in section 3 and the tables in the appendices seem to suggest this distinction is only being done in the code generation tasks. This isn't a bad thing per se (indeed this distinction makes the most sense in the context of code generation), but the inconsistency of the variations, added to the sheer number of them, makes it harder to form a coherent and compelling picture of the results. In a similar vein, I also find that I'm pretty confused as to which experiments included results with GPT-3.5 as well as GPT-4o, vs which ones only included results with GPT-3.5.

4. Confidence intervals could be calculated and included in the bar charts. The results seem to be generally close enough that this could matter a fair amount.

**Questions:**

1. How confident are you that the main results are not spurious? By which I mean: how likely does it seem that the results would generalize with more numerous and systematic variations on each problem aspect studied (e.g. if there were more systematic variation within each "multi-agent system structure", each "task category", etc)? What evidence are you relying on for your assessment?

2. How much iteration was done in the prompting of the systems? It seems plausible to me that many of the observed shortcomings of the multi-agent systems, the malicious agent simulators (e.g. the relative inability of AutoTransform to decrease performance on Translation and TextEval), and the defense methods may be attributed to insufficiently refining of the methods.

3. Figure 3b includes results from a single GPT-3.5 agent. Are all other agent systems here exclusively using GPT-3.5, or is this including results with GPT-4o? The text doesn't make this clear. I'm guessing they all just use GPT-3.5, in which case it's all fine, but if not, then this would raise additional questions. In particular, it would seem that the simple baseline of a single GPT-4o agent would beat the multi-agent systems, and the rest of the investigation would be a bit closer to moot.

4. A related question to the above: the fact that code generation as a task is more susceptible to sabotage by malicious agents seems surprising to me, since it is the most verifiable of the tasks (running the code provides a source of truth for its functionality that does not depend on trust in the specific agents). This is another example of my feeling that simple baselines can possibly beat many of the setups described. Is there a reason why the agents were not able to verify the code by running it?

**Details Of Ethics Concerns:**

I don't think this paper is net harmful and I think that this type of work is important for building safer systems. I would not like to see this type of work be slowed down due to ethical concerns (I think that would be counterproductive to ethics). But it is the case that this paper presents potentially harmful methodologies, so I'm flagging it for further review.

---

> ### Author Response · Authors · 2024-11-25
> **Official Response (1/n)**
>
> We deeply appreciate your efforts in reviewing and your recognition of the originality and significance of our paper. Your feedback has significantly improved our paper. In the response, we address your concerns one by one.
>
> > My main concern is that this paper attempts to do too much and is not sufficiently focused. Between the different multi-agent structures, different tasks, different attacks, different types of error rates, different error types, and different defenses, there are too many variables, each investigation ends up with limited depth, and the overall picture ends up not fully compelling. As a result of this ambition, details in the subquestions appear insufficiently investigated. For example, when discussing different multi-agent structures, I hoped to find more details about the structures and their dynamics, and more possible instantiations of each high-level structure (or, more systematic variation in the instantiations considered). Then the rest of the setup could be simplified (for example, it could focus on a single task category such as code generation, again possibly with more than one instantiation of the task category). The resulting claim would have to be more modest -- for example, it would pertain only to code generation and not any task -- but it would be much more strongly substantiated. As it stands (especially given the results do not seem particularly extreme), I'm left wondering if it is really the case that hierarchical structures are more resilient, or if the results apply only to the specific hierarchical systems tested and whether there is something particular about each of them that led to the results. I'm left unconvinced that the main claims of the paper are true.
>
> Thank you for your thoughtful feedback. We recognize that the limited number of system instantiations in each structure may constrain the depth of our analysis. Our aim in this work was to **offer a broad perspective on the factors influencing resilience in multi-agent systems** as a foundation for future exploration. While this study focuses on providing a comprehensive overview, we acknowledge the importance of deeper investigation into specific structures and their dynamics. In future work, we plan to expand our evaluations to include more systematic variations and additional instantiations, such as AutoGen, to further substantiate our findings and address the concerns you raised.
>
> > A related general concern is that the lack of systematic ablations or closer analyses of the specific results made me quite doubtful of them. I think many additional experiments could have been very illustrative. For example, in the investigation of the defense methods, I was left wondering how much the relation between the defense methods and the attack methods mattered. I find it plausible that these "defense methods" are just generally useful enhancements to the multi-agent systems, and was interested in seeing results on the multi-agent systems with the "defense methods" even without the attack in place. In that case, the discussion of these results would be a bit different.
>
> Thank you for your thoughtful suggestion. To address this concern, we conducted additional experiments to evaluate (1) the **combination** of Challenger and Inspector, (2) performance in **“No Attack”** scenarios, and (3) defense against **AutoTransform** attacks. The expanded results, including the previously reported Camel and Self-collab experiments, are summarized below:
>
> | Self-collab | No Defense | Challenger | Inspector | Challenger + Inspector |
> |---|---|---|---|---|
> | No Attack | 76.22 | 74.56 | 76.39 | 76.83 |
> | AutoTransform | 43.29 | 70.73 | 74.40 | 75.00 |
> | AutoInject | 40.85 | 71.95 | 67.68 | 73.78 |
>
> | Camel | No Defense | Challenger | Inspector | Challenger + Inspector |
> |---|---|---|---|---|
> | No Attack | 62.20 | 62.23 | 61.03 | 63.79 |
> | AutoTransform | 32.46 | 43.50 | 41.75 | 48.70 |
> | AutoInject | 29.27 | 40.24 | 44.16 | 48.64 |
>
> The results show that while our defense methods significantly enhance resilience to attacks, they **provide only marginal improvements in “No Attack” scenarios**. This indicates that their utility is primarily in mitigating adversarial challenges, rather than as general system enhancements.

---

> > ### Author Response · Authors · 2024-11-25
> > **Official Response (2/n)**
> >
> > > I found the combination of experiments a bit confusing and unclear. Part of this is directly downstream of point 1 (there are too many axes of variation), but it is also the case that the different factors are, it seems, inconsistently varied. For example, the introduction of "Error Types" in section 3 and the tables in the appendices seem to suggest this distinction is only being done in the code generation tasks. This isn't a bad thing per se (indeed this distinction makes the most sense in the context of code generation), but the inconsistency of the variations, added to the sheer number of them, makes it harder to form a coherent and compelling picture of the results. In a similar vein, I also find that I'm pretty confused as to which experiments included results with GPT-3.5 as well as GPT-4o, vs which ones only included results with GPT-3.5.
> >
> > Thank you for your detailed feedback. To clarify the experimental design and address concerns about inconsistency, we have added a summary table for better visualization of the setup:
> >
> > | Systems | Structure | Tasks | N. of Agents | Final Agent | Malicious Agent |
> > |---|---|---|---|---|---|
> > | MetaGPT | Linear | Code | 5 | Test Engineer | Code Engineer |
> > | Self-collab | Linear | Code | 2-5 | Tester | Coder |
> > | Camel | Flat | All | 2 | User | Assistant |
> > | SPP | Flat | Code | 3 | AI Assistant | Python Programmer |
> > | MAD | Hierarchical | All | 3 | Judge | debater |
> > | AgentVerse | Hierarchical | All | 4 | Critic | Solver |
> >
> > **All experiments in the main text are conducted with GPT-3.5**. Additional experiments using GPT-4o are presented in Appendix A to provide broader insights. For completeness, we have also included results with a state-of-the-art open-source model, **LLaMA-3.1-70B-Instruct**, as part of our rebuttal. Below are the results summarized across system structures:
> >
> > | LLaMA-3.1-70B-Instruct | Linear | Flat | Hierarchical |
> > |---|---|---|---|
> > | No Attack | 73.78 | 76.83 | 76.15 |
> > | AutoTransform | 11.90 | 39.03 | 66.96 |
> > | AutoInject | 38.72 | 36.59 | 55.64 |
> >
> > > Confidence intervals could be calculated and included in the bar charts. The results seem to be generally close enough that this could matter a fair amount.
> > > How much iteration was done in the prompting of the systems? It seems plausible to me that many of the observed shortcomings of the multi-agent systems, the malicious agent simulators (e.g. the relative inability of AutoTransform to decrease performance on Translation and TextEval), and the defense methods may be attributed to insufficiently refining of the methods.
> >
> > Thank you for this valuable feedback. While we agree that confidence intervals could provide additional insights, the extensive scope of experiments in this study presents significant computational and financial constraints for running tasks multiple times to generate these intervals. However, we believe the large number of test cases included in each task offers a robust basis for statistically meaningful results. Furthermore, the **observed trends are consistent across a diverse set of scenarios**, supporting the reliability of our findings.
> >
> > > How confident are you that the main results are not spurious? By which I mean: how likely does it seem that the results would generalize with more numerous and systematic variations on each problem aspect studied (e.g. if there were more systematic variation within each "multi-agent system structure", each "task category", etc)? What evidence are you relying on for your assessment?
> >
> > Thank you for your insightful comment. We acknowledge the vast diversity in real-world tasks, roles, prompts, and multi-agent system structures. To address this, we carefully selected **six widely-used multi-agent frameworks** and evaluated them across **four representative downstream tasks**. By employing three backbone models—**GPT-3.5, GPT-4o, and LLaMA-3-8B-Instruct**—we ensured robustness in our analysis. Notably, GPT-4o, despite being a stronger model, corroborated the findings from GPT-3.5, reinforcing the consistency of our results. This alignment across models suggests that our conclusions are generalizable, and we anticipate they will remain relevant as LLMs continue to evolve.

---

> > > ### Author Response · Authors · 2024-11-25
> > > **Official Response (3/n)**
> > >
> > > > Figure 3b includes results from a single GPT-3.5 agent. Are all other agent systems here exclusively using GPT-3.5, or is this including results with GPT-4o? The text doesn't make this clear. I'm guessing they all just use GPT-3.5, in which case it's all fine, but if not, then this would raise additional questions. In particular, it would seem that the simple baseline of a single GPT-4o agent would beat the multi-agent systems, and the rest of the investigation would be a bit closer to moot.
> > >
> > > We appreciate the reviewer’s observation. To clarify, all experiments in Fig. 3 (a) and (b) utilize GPT-3.5 for consistency and comparability. Similarly, all experiments in Fig. 9 (a) and (b) are conducted using GPT-4o. **Our main conclusions remain consistent across both models**: hierarchical structures demonstrate superior performance; rigorous tasks exhibit greater sensitivity to malicious agents; and systems like MAD and Camel also show notable performance improvements.
> > >
> > > > A related question to the above: the fact that code generation as a task is more susceptible to sabotage by malicious agents seems surprising to me, since it is the most verifiable of the tasks (running the code provides a source of truth for its functionality that does not depend on trust in the specific agents). This is another example of my feeling that simple baselines can possibly beat many of the setups described. Is there a reason why the agents were not able to verify the code by running it?
> > >
> > > We appreciate this insightful observation. While integrating an external interpreter or execution tool can indeed assist in detecting syntactic errors, it has **limitations in addressing deeper semantic issues**. For instance, systems like **Self-collab employ tools to verify code correctness but remain vulnerable to semantic errors** introduced by malicious agents. This limitation highlights the need for robust mechanisms beyond simple execution-based verification, as these cannot capture the nuanced sabotage strategies we investigate.
> > >
> > > > I don't think this paper is net harmful and I think that this type of work is important for building safer systems. I would not like to see this type of work be slowed down due to ethical concerns (I think that would be counterproductive to ethics). But it is the case that this paper presents potentially harmful methodologies, so I'm flagging it for further review.
> > >
> > > Thank you for your thoughtful feedback. We acknowledge the potential risks associated with the methodologies presented in our work. To address these concerns, we propose two defense **mechanisms—the Challenger and the Inspector—and their combination**. These methods have been rigorously evaluated and demonstrate significant effectiveness in mitigating the influence of malicious agents, as evidenced by the results below:
> > >
> > > | Self-collab | No Defense | Challenger | Inspector | Challenger + Inspector |
> > > |---|---|---|---|---|
> > > | No Attack | 76.22 | 74.56 | 76.39 | 76.83 |
> > > | AutoTransform | 43.29 | 70.73 | 74.40 | 75.00 |
> > > | AutoInject | 40.85 | 71.95 | 67.68 | 73.78 |
> > >
> > > | Camel | No Defense | Challenger | Inspector | Challenger + Inspector |
> > > |---|---|---|---|---|
> > > | No Attack | 62.20 | 62.23 | 61.03 | 63.79 |
> > > | AutoTransform | 32.46 | 43.50 | 41.75 | 48.70 |
> > > | AutoInject | 29.27 | 40.24 | 44.16 | 48.64 |
> > >
> > > These results illustrate that the proposed methods effectively mitigate potential harm while maintaining system performance, reinforcing the value of this research for building safer systems.

---

> > > > ### Comment · Reviewer_Q1ar · 2024-11-26
> > > >
> > > > I thank the authors for the detailed responses. I feel they added significant clarity and I felt my concerns were heard. I am especially thankful that the authors ran the experiment of applying the defenses in the "No attack" scenario -- the results are interesting and compelling.
> > > >
> > > > Still, the authors' response has not allayed my primary concern, which is that the paper attempts to do too much. While the authors' response adds significant clarity, my main point is that these clarifications, in a sense, should not be needed. I feel somewhat in analogy to a code reviewer who comments that some code is unclear, and receives in response an explanation in text of what the code does. It is helpful for my understanding, but it does not change my feelings about the quality of the code.
> > > >
> > > > That said, the authors' response has significantly increased my confidence in the authors' overall line of work, and I'm optimistic that an updated, more focused version of this work, could be quite compelling.

---

> > > > > ### Author Response · Authors · 2024-11-28
> > > > >
> > > > > Thank you for reading our response. We appreciate and are encouraged that you find our “No Attack” experiment helpful!
> > > > >
> > > > > We understand your concern about the breadth of the paper. However, we believe that a broader scope is essential at this preliminary stage to derive generalizable insights. A highly focused setting, while providing deeper insights into a specific scenario, risks limiting the applicability of the findings. For instance, if we were to conclude that a hierarchical structure performs best for code generation with 20% syntactic errors, such a result **might not hold for other tasks, such as math problem solving, or with different levels of error, like 40% or 60%**. By maintaining a broader approach, we aim to establish foundational, high-level conclusions that can guide future, more targeted investigations. This broader perspective lays the groundwork for deeper, scenario-specific studies moving forward.

---

### Official Review · Reviewer_QzAB · 2024-11-05

**Soundness:** 3
**Presentation:** 3
**Contribution:** 2
**Rating:** 5
**Confidence:** 3

**Summary:**

The paper experimentally examine the resilience of LLM-based multi-agent systems with various system architectures, when malicious agent presents. The work designs two methods to simulate malicious agents, and design two corresponding defense methods. The paper designs several experiments to examine how different types of system perform on several downstream tasks given various degree of errors injected by malicious agents.

**Strengths:**

The research direction is interesting and of great significance. The presentation is overall clear, not hard to follow. Various experiments are designed and several interesting observations are provided.

**Weaknesses:**

The overall weaknesses concern with the contributions, presentation of details, and the experiment
1. The design of malicious agents can be rather trivial and heuristic, and lacking representation guarantees.

    * The proposed methods can be restricted and there is no *guarantee* whether the proposed two methods reflect (at least the majority types of) real-world attacks. As the author pointed out, AutoTransform is convenient, yet hard to analyze. This inherently does not align with the objectives of the paper, because this method provide minimal added insights over AutoInject. It is thus not clear why an LLM-based approach is necessary and considered one of the contributions. A more principled automatic approach that attempts to capture different types of attacks can be interesting to explore.

    * While AutoInject seems more principal, whether $P_m$ and $P_e$, the degree of error injected on the input side represents a good error rate metric is doubted, because even injecting the same number of errors per line can lead to different output behavior. For instance, in AutoInject, both injecting error only on a single line of code `while b`, changing it to (1) `while b>=0` or (2) `while True` leads completely different results. In the latter case, if the agent running the code has no mechanism to jump out of infinite loop, this leads to catastrophic propogation of error to the entire system. In this example, it is clear the error in case (2) can be more dangerous, yet the provided error rate metric seems too trivial to capture it.

2. The specific research questions seem shallow, the presentation of experiment results are not clear, and for certain interesting observed  phenomenon, the provided insights seem limited.

    * The paper only discussed the observed phenomenon, and do not seem to deepen the research area by providing more insights how to use the consequences of these observations to design better resilient system. For instance, in certain systems, it may be inevitable to choose a linear architecture. Given these observations, can we join a proposed defense method to make it more closely resemble a hierarchical system, so as to demonstrate the usefulness and significance of the observed results?

    * Similarly, for the surprising observation that "introduced errors can cause performance increase", we only see discussion up to the reasons, but not how this result leads any designing insights. In particular, if agents are already capable of double checking the results and identifying the injected errors, how the proposed defense methods, which are designed to challenge the results of others, provide additional help over such tasks?

    * The experiment settings are very vaguely presented. It is not clear which agent is malicious, which agent output the final results, and which task is used to evaluate different architectures. Or the experiment results represent the average performance under all different settings. It is also not clear how many agents are there, and thus not clear if the conclusion holds only for a small-scaled system, or can be generalized to more complicated systems.

    * Figure 3 is very poorly plotted. The title says the figures demonstrate "performance drops" thus one would think the y-axis represents the percentage drops compared to an intact system. However, it seems the y-axis corresponds to the absolute performance metrics. Are different tasks have the same metric? If not, then why does it make sense to compare different tasks on the same scale? It is also not immediately clear what "Vanilla" refers to, as they only appear in the plots.


3. While it is claimed in Abstract that the paper investigates the question of how we can increase system resilience to defend against malicious agents, the paper has limited discussion on this. The paper provides no definitive answer whether the proposed method achieves consistent performance gain in various scenarios, and cannot guarantee performance over more realistic scenarios.

**Questions:**

1. You mentioned that the proposed defense methods (Challenger and Inspector) correspond to the two simulation methods (AutoTransform and AutoInject). It is then natural to explore how well these defense methods fix the corresponding type of malicious agent. Do you believe, if e.g., a malicious agent is due to AutoInject, then the Inspector defense should work consistently better?

---

> ### Author Response · Authors · 2024-11-25
> **Official Response (1/n)**
>
> We deeply appreciate your efforts in reviewing and your recognition of our extensive experiments and the clear presentation of our paper. Your feedback has significantly improved our paper. In the response, we address your concerns one by one.
>
> > The proposed methods can be restricted and there is no guarantee whether the proposed two methods reflect (at least the majority types of) real-world attacks.
>
> We appreciate the reviewer’s insightful comment. We acknowledge the diversity of real-world attack types, including DDoS and MITM. Our study focuses on simulating highly stealthy attacks—those that appear benign at first glance or are difficult for non-experts to detect. Specifically, we target scenarios where malicious content, such as a deliberately incorrect line of code, **is embedded within seemingly innocuous messages**. This approach aims to reflect a critical subset of real-world attacks that prioritize subtlety and evasion.
>
> > As the author pointed out, AutoTransform is convenient, yet hard to analyze. This inherently does not align with the objectives of the paper, because this method provide minimal added insights over AutoInject. It is thus not clear why an LLM-based approach is necessary and considered one of the contributions. A more principled automatic approach that attempts to capture different types of attacks can be interesting to explore.
>
> Thank you for the insightful feedback. While we acknowledge that AutoTransform's reliance on LLM-generated outputs introduces challenges in control and analysis, its inclusion addresses a key limitation of AutoInject: **the inability to replicate the inherent behaviors of LLMs**. AutoInject-generated errors may not accurately reflect errors that LLMs themselves might produce in real-world scenarios. Furthermore, AutoTransform allows us to investigate **whether LLMs can be guided to generate errors that are sufficiently covert to obfuscate malicious intent**. This exploration aligns with our broader objective of understanding LLM vulnerabilities and highlights the necessity of a goal-driven approach to address complex attack dynamics.
>
> > While AutoInject seems more principal, whether P_m and P_e, the degree of error injected on the input side represents a good error rate metric is doubted, because even injecting the same number of errors per line can lead to different output behavior. For instance, in AutoInject, both injecting error only on a single line of code while b, changing it to (1) while b>=0 or (2) while True leads completely different results. In the latter case, if the agent running the code has no mechanism to jump out of infinite loop, this leads to catastrophic propogation of error to the entire system. In this example, it is clear the error in case (2) can be more dangerous, yet the provided error rate metric seems too trivial to capture it.
>
> Thank you for highlighting this important consideration. We agree that different types of errors can have varied impacts on program functionality and may pose different challenges for detection. While our current metric—errors per line of code ($P_m$ and $P_e$)—is relatively simple, it provides a **practical and interpretable approach akin to how lines of code are widely used as a baseline metric in software development** despite the existence of more nuanced alternatives.
>
> To address your concern regarding **error diversity and severity**, we further analyzed the distribution of error types generated by AutoInject. The errors span across seven distinct categories, as detailed below, ensuring diversity in the types of faults injected and reducing the bias of any single category dominating the results:
>
> | Category Name | Description | Count |
> |---|---|---|
> | Logical Errors | Errors in logical operations, such as incorrect operators or inverted logic. | 12 |
> | Indexing and Range Errors | Issues with boundary conditions or off-by-one indexing. | 23 |
> | Mathematical Errors | Errors in calculations or numerical processing. | 20 |
> | Output and Formatting | Issues with producing or formatting expected output. | 9 |
> | Initialization Errors | Problems with starting values or incorrect initialization. | 4 |
> | Infinite Loops | Errors causing unintended infinite execution loops. | 6 |
> | Runtime Invocation Issues | Errors in function calls or runtime handling. | 6 |
>
> By incorporating a diverse range of errors and generating them at scale, AutoInject effectively captures the broad spectrum of fault types, mitigating the risk that specific critical cases—like infinite loops—are overlooked. This approach ensures that **the reported error metrics, while simple, remain robust and representative of diverse error scenarios.**

---

> > ### Author Response · Authors · 2024-11-25
> > **Official Response (2/n)**
> >
> > > The paper only discussed the observed phenomenon, and do not seem to deepen the research area by providing more insights how to use the consequences of these observations to design better resilient system. For instance, in certain systems, it may be inevitable to choose a linear architecture. Given these observations, can we join a proposed defense method to make it more closely resemble a hierarchical system, so as to demonstrate the usefulness and significance of the observed results?
> >
> > Thank you for highlighting the need to demonstrate the practical implications of our observations. To address this, we evaluated the effectiveness of our proposed defense methods—**Challenger, Inspector, and their combination**—within a linear system, Self-collab. The results below illustrate **the improved resilience against both AutoTransform and AutoInject attacks:**
> >
> > | Self-collab | No Defense | Challenger | Inspector | Challenger + Inspector |
> > |---|---|---|---|---|
> > | No Attack | 76.22 | 74.56 | 76.39 | 76.83 |
> > | AutoTransform | 43.29 | 70.73 | 74.40 | 75.00 |
> > | AutoInject | 40.85 | 71.95 | 67.68 | 73.78 |
> >
> > As shown, integrating our defense methods significantly enhances the robustness of the linear system. This demonstrates that our findings are not only theoretically valuable but also practically applicable in designing more resilient systems, even when constrained to a linear architecture.
> >
> > > Similarly, for the surprising observation that "introduced errors can cause performance increase", we only see discussion up to the reasons, but not how this result leads any designing insights. In particular, if agents are already capable of double checking the results and identifying the injected errors, how the proposed defense methods, which are designed to challenge the results of others, provide additional help over such tasks?
> >
> > We appreciate the reviewer’s insightful comment. The Challenger contributes to system resilience by **explicitly instructing agents to critically evaluate and challenge others’ results, an initiative they might not otherwise undertake independently.** As demonstrated in the case study (Fig. 6(a), Page 8), a single line of erroneous code is insufficient for detection by other agents. However, when AutoInject introduces additional erroneous lines, the agents identify the discrepancy and prompt the coder to refine its results. This highlights a key design insight: in multi-agent debate frameworks, **intentionally injecting errors—particularly those outside LLMs’ standard distribution—can foster divergent thinking,** encourage thorough verification, and ultimately lead to more refined and agreed-upon results.
> >
> > > The experiment settings are very vaguely presented. It is not clear which agent is malicious, which agent output the final results, and which task is used to evaluate different architectures. Or the experiment results represent the average performance under all different settings. It is also not clear how many agents are there, and thus not clear if the conclusion holds only for a small-scaled system, or can be generalized to more complicated systems.
> >
> > We thank the reviewer for highlighting the need for clarity in our experimental settings. To address this concern, we have included a detailed table summarizing the experimental configurations:
> >
> > | Systems | Structure | Tasks | N. of Agents | Final Agent | Malicious Agent |
> > |---|---|---|---|---|---|
> > | MetaGPT | Linear | Code | 5 | Test Engineer | Code Engineer |
> > | Self-collab | Linear | Code | 2-5 | Tester | Coder |
> > | Camel | Flat | All | 2 | User | Assistant |
> > | SPP | Flat | Code | 3 | AI Assistant | Python Programmer |
> > | MAD | Hierarchical | All | 3 | Judge | debater |
> > | AgentVerse | Hierarchical | All | 4 | Critic | Solver |
> >
> > This table provides clarity on the number of agents, the tasks evaluated, the malicious agent setup, and the agent responsible for the final output. We have also clarified whether results represent average performance across multiple configurations.
> >
> > We acknowledge that the multi-agent systems analyzed in this work are relatively small in scale (<6 agents). However, the majority of contemporary research in multi-agent systems employs a limited number of agents. Therefore, **we believe that our conclusions are generalizable to most frameworks, such as AutoGen**, and can serve as a foundation for scaling up to larger systems in future work.

---

> > > ### Author Response · Authors · 2024-11-25
> > > **Official Response (3/n)**
> > >
> > > > Figure 3 is very poorly plotted. The title says the figures demonstrate "performance drops" thus one would think the y-axis represents the percentage drops compared to an intact system. However, it seems the y-axis corresponds to the absolute performance metrics.
> > >
> > > Thank you for highlighting this issue. We have revised the caption of Figure 3 (as well as Figures 5 and 9) to clarify the representation. The new caption explicitly describes the y-axis as **absolute performance metrics**, ensuring alignment with the data presented.
> > >
> > > > Are different tasks have the same metric? If not, then why does it make sense to compare different tasks on the same scale? It is also not immediately clear what "Vanilla" refers to, as they only appear in the plots.
> > >
> > > All four tasks use accuracy as the evaluation metric, **ranging from 0 to 1.** For the translation task, accuracy specifically measures n-gram precision between the translated text and the reference text. The term "Vanilla" denotes **a baseline scenario where no attack or defense methods are applied** to the system.
> > >
> > > > While it is claimed in Abstract that the paper investigates the question of how we can increase system resilience to defend against malicious agents, the paper has limited discussion on this. The paper provides no definitive answer whether the proposed method achieves consistent performance gain in various scenarios, and cannot guarantee performance over more realistic scenarios.
> > >
> > > Thank you for your thoughtful suggestion. To address this concern, we conducted additional experiments to evaluate (1) the **combination** of Challenger and Inspector, (2) performance in **“No Attack”** scenarios, and (3) defense against **AutoTransform** attacks. The expanded results, including the previously reported Camel and Self-collab experiments, are summarized below:
> > >
> > > | Self-collab | No Defense | Challenger | Inspector | Challenger + Inspector |
> > > |---|---|---|---|---|
> > > | No Attack | 76.22 | 74.56 | 76.39 | 76.83 |
> > > | AutoTransform | 43.29 | 70.73 | 74.40 | 75.00 |
> > > | AutoInject | 40.85 | 71.95 | 67.68 | 73.78 |
> > >
> > > | Camel | No Defense | Challenger | Inspector | Challenger + Inspector |
> > > |---|---|---|---|---|
> > > | No Attack | 62.20 | 62.23 | 61.03 | 63.79 |
> > > | AutoTransform | 32.46 | 43.50 | 41.75 | 48.70 |
> > > | AutoInject | 29.27 | 40.24 | 44.16 | 48.64 |
> > >
> > > These results demonstrate that **combining the Challenger and Inspector defenses consistently improves system performance** under malicious attacks across various scenarios. We recommend adopting such a multi-agent defense strategy to enhance system resilience.
> > >
> > > > You mentioned that the proposed defense methods (Challenger and Inspector) correspond to the two simulation methods (AutoTransform and AutoInject). It is then natural to explore how well these defense methods fix the corresponding type of malicious agent. Do you believe, if e.g., a malicious agent is due to AutoInject, then the Inspector defense should work consistently better?
> > >
> > > Thank you for the insightful comment. Our experiments with AutoTransform, as detailed **in the tables referenced in the previous question**, show that Challenger outperforms Inspector against AutoTransform, while Inspector is more effective against AutoInject in the Camel system. Interestingly, this trend is reversed in the Self-collab system. These findings suggest that **the effectiveness of a defense method is influenced more by the system architecture** than by the type of attack method employed.

---

> > ### Comment · Reviewer_QzAB · 2024-11-27
> >
> > I appreciate the authors for their thoroughly addressing my concerns and for their efforts in preparing more results. I believe that the highlighted messages in your responses are all crucial to demonstrate the significance of your work and clarify a lot of the confusions and concerns. I would love to see the rebuttal properly integrated and better presented in your manuscript.
> >
> > At the same time, I agree with the comments made by Reviewer Q1ar that "the paper attempts to do too much." This is consistent with one of my earlier concerns that you want to include more insights of the key observations, rather than vaguely presenting everything.
> > Thus, I would suggest that the authors to attempt a critical restructuring of the paper such that it can better clarify the settings and showcase the key takeaways, while defering some of the less interesting findings to the appendices.

---

> > > ### Author Response · Authors · 2024-11-28
> > >
> > > We appreciate your feedback and acknowledge the concern regarding the breadth of the paper. Our intention in adopting a broader scope is to establish foundational insights that are generalizable across various settings, rather than limiting the analysis to highly specific scenarios. For example, focusing exclusively on a hierarchical structure for code generation with 20% syntactic errors might yield conclusions that **do not extend to other tasks, such as mathematical problem solving, or to cases with different error levels (e.g., 40% or 60%)**. By presenting a broader analysis, we aim to provide a versatile framework that can guide future studies into more focused and scenario-specific investigations. To address your suggestion, we will restructure the paper to better highlight the key takeaways and defer some of the less critical findings to the appendices.

---

### Author Response · Authors · 2024-11-25
**Author Response Period Summary**

We deeply thank all reviewers for their time, efforts, and insightful feedback. Their suggestions have greatly improved our work. We are particularly encouraged by reviewer’s recognition of:

- **Interesting and significant research direction** (QzAB, Q1ar, jrv1, 3j9A)
- **Clear and structured presentation** (QzAB, Q1ar, jrv1)
- **Comprehensive and robust experimentation** (QzAB, 54qu, 3j9A)
- **Insightful findings and observations** (QzAB, Q1ar, 54qu, 3j9A)

During the author response period, we have considered the constructive suggestions provided and made several significant improvements our manuscript, including:

- Analysis of **error diversity and error types** in AutoInject (QzAB, 54qu)
- More detailed evaluations of **defense methods** (QzAB, Q1ar, jrv1, 3j9A)
- Results with **LLaMA-3.1** (Q1ar, jrv1, 54qu)
- Related work in **distributed systems** (jrv1, 3j9A)
- **Improved presentation** (QzAB, Q1ar, 54qu, jrv1, 3j9A)

Once again, we sincerely thank the reviewers for their thoughtful suggestions and valuable contributions to enhancing our paper.

---

### Meta-Review · Area_Chair_W7xr · 2024-12-19

**Metareview:**

The reviewers acknowledged that the paper tackles an important, timely question about the robustness of various LLM-based multi-agent systems, and provides interesting connections between the system's structure and its resilience. However, the reviewers pointed out several weaknesses and shared concerns related to unclear evaluation setup, lack of systematic ablation experiments,  and limited discussion regarding the existing literature on the resilience of non-LLM-based multi-agent systems. We want to thank the authors for their detailed responses. Based on the raised concerns and follow-up discussions, unfortunately, the final decision is a rejection. Nevertheless, this is exciting and potentially impactful work, and we encourage the authors to incorporate the reviewers' feedback when preparing a future revision of the paper.

**Additional Comments On Reviewer Discussion:**

The reviewers pointed out several weaknesses and shared concerns related to unclear evaluation setup, lack of systematic ablation experiments,  and limited discussion regarding the existing literature on the resilience of non-LLM-based multi-agent systems. A majority of the reviewers support a rejection decision and agree that the paper is not yet ready for acceptance.

---

### Decision · Program_Chairs · 2025-01-22

Reject